# Real-Time Estimation of Temperature Time Derivative in Inertial Measurement Unit by Finite-Impulse-Response Exponential Regression on Updates

**DOI:** 10.3390/s20051299

**Published:** 2020-02-27

**Authors:** Alexander Kozlov, Ilya Tarygin

**Affiliations:** Lomonosov Moscow State University, Faculty of Mechanics and Mathematics, Navigation and Control Lab. Leninskiye Gory, Main Building, 119991 Moscow, Russia; i.tarygin@gmail.com

**Keywords:** temperature sensor, temperature time derivative, numerical differentiation, inertial measurement unit

## Abstract

We present a filtering technique that allows estimating the time derivative of slowly changing temperature measured via quantized sensor output in real time. Due to quantization, the output may appear constant for several minutes in a row with the temperature actually changing over time. Another issue is that measurement errors do not represent any kind of white noise. Being typically the case in high-grade inertial navigation systems, these phenomena amid slow variations of temperature prevent any kind of straightforward assessment of its time derivative, which is required for compensating hysteresis-like thermal effects in inertial sensors. The method is based on a short-term temperature prediction represented by an exponentially decaying function, and on the finite-impulse-response Kalman filtering in its numerically stable square-root form, employed for estimating model parameters in real time. Instead of using all of the measurements, the estimation involves only those received when quantized sensor output is updated. We compare the technique against both an ordinary averaging numerical differentiator and a conventional Kalman filter, over a set of real samples recorded from the inertial unit.

## 1. Introduction

In inertial measurement units (IMU) of navigation grade, thermal effects usually have a substantial impact on inertial sensor errors. Among others, the output of, e.g., a fiber optical gyroscope, may depend not only on the temperature itself, but on its rate of change as well. Examples of signal deviation under a non-zero temperature time derivative, including hysteresis, may be found in the literature [1]. To compensate (see [2]) for the described effects, apart from an appropriate model, we need an actual value of the rate at which the temperature changes. Obtaining this value, however, becomes a challenge in such devices, due to the combination of several issues commonly inherent to inertial systems of navigation grade.

The first concern is the quantization of measurements provided by conventional temperature sensors, which rarely have their resolution better than 0.05–0.2 °C. In terms of accuracy, this resolution perfectly suits all the needs for measuring temperature itself, but appears to be far from that required for the estimation of temperature time derivative in any ordinary manner. The second issue is a rather wide range of rates the temperature variations can have, going from nearly zero to several degrees per minute. The third one lies in the assumptions typically laid on the data to which conventional digital filters are applied. Neither measurement errors have properties close to that of white noise, nor the thermal process itself conforms well to describing in terms of frequency domains of its periodic components.

We aim to assess the time derivative of temperature with an error within a degree per hour, which corresponds to changing the temperature sensor reading by one single quantization step in the whole course of roughly 5 min. Meanwhile, since the derived quantity is to be used in high-frequency inertial computations at approximately 100–500 Hertz, it should be available at every single epoch on that frequency. Figure 1 illustrates the point better than calculating in numbers, that a straightforward numerical differentiation of a measured signal, without any additional considerations, would yield nowhere near the actual temperature time derivative. The data shown come from a temperature sensor installed inside a ring laser gyroscope. In the Figure, there are periods of constant output for about 3–7 min (see from minute 0 to 7, then from 8 to 11, etc.), suggesting zero derivative, then periods of signal flickering back and forth (from 7 to 8, 11 to 12, etc.) in the top left inset with an average temperature rate of roughly 0.5 °C/hr, and periods of fairly rapid temperature change of about 5 °C per hour, as shown in the bottom right inset. The example displayed here has been recorded during a self-heating of the system in a test environment. Real in-flight variations may grow over time by up to one order of magnitude faster.

Under these conditions, no conventional digital differentiating filter will properly work both during the periods when the signal is essentially constant but the actual temperature is not, during the periods of flickering, and during the periods in which the temperature changes rather quickly at a varying rate. In general, the larger time constant the filter has, the greater phase shift it generates in its output. It may seem plausible to cherry-pick parameter values for a conventional filter for a particular data set, yet it still appears to be impossible to cover all scenarios happening in real practice. Therefore, some kind of adaptive filter should perform here.

In this paper, we suggest using a technique relying on the physical nature of the internal thermal processes taking place inside the IMU, employing Kalman filtering [3] to optimize their model parameters in real time. Although there are solutions in the literature for smooth temperature estimation [4] out of noisy measurements, none of them address the quantized nature of temperature sensor output under slow thermal variations in a high-grade IMU.

## 2. Background Models and Algorithms

### 2.1. Thermal Behavior

Prior to coming to real time filtering, let us first consider the background description of thermal processes in inertial measurement units [5]. Though emerging inside a complex structure of different kinds of instrumentation and thus being extremely difficult to model, thermal processes in general tend to obey the well-known heat equation, written here in polar 3-D coordinates {r,φ,ψ} and normalized:(1)$$\tau =\tau \left(t,r,\varphi ,\psi \right),\;\frac{\partial \tau }{\partial t}=\frac{1}{r^{2}}\frac{\partial }{\partial r}\;\;\left(r^{2}\frac{\partial \tau }{\partial r}\right)+\frac{1}{r^{2}\sin\varphi }\frac{\partial }{\partial \varphi }\;\;\left(\sin\varphi \frac{\partial \tau }{\partial \varphi }\right)+\frac{1}{r^{2}\sin^{2}\varphi }\frac{\partial^{2}\tau }{\partial \psi^{2}}+\delta \left(r,\varphi ,\psi \right),$$
where τ is the normalized temperature, *t* stands for time, and δ(r,φ,ψ) denotes the heat source function. The real conditions under which the thermal processes take place inside the inertial measurement unit are, of course, a matter for a special discussion on their own. More than that, to even start speaking of a solution, Equation (Equation 1) requires boundary and initial conditions to be added, which are always unknown in real practice. Ignoring the above-mentioned complications left for the whole separate branch of mathematics, the best of what we can practically obtain from the general theory implies that for “sufficiently” stationary heat sources and boundary effects inside a limited volume of the “sufficiently” uniform isotropic medium, the general solution mostly appearing in the literature takes the following form, expressed as a series of spatially and temporally sharpening terms:$$\tau \left(t,r\right)=a\left(r\right)+\sum_{m=1}^{\infty }\left(\frac{b_{m}}{r}\sin\sqrt{m}\frac{r}{r_{0}}+\frac{c_{m}}{r}\cos\sqrt{m}\frac{r}{r_{0}}\right)e^{-mt/T_{0}},$$
where a(r) is defined by heat sources, $T_{0}$ denotes a time constant, and $r_{0}$ stands for a linear spatial scale, with other parameters depending on initial and boundary conditions. Indeed, thermal processes in rather massive objects with barely moving parts and no rapid changes in temperature, like inertial measurement units of navigation grade, are commonly accepted to have some sort of exponential decay in time. Additionally, for these types of devices we may reasonably expect that more “rapid” higher terms in the expansion will contribute less to temperature variation, and thus we will stop adding them from some point on. This clearly appears to be the case at least for self-heating, so that for the above demonstrated sample an approximation of six exponential terms fits, on a global scale, the sensor readings perfectly within the limits of practical application for the entire record of more than three hours, as shown in Figure 2.

For further generalization, since real heat sources and boundary conditions may vary with time, we will accept the form of temperature model function at the measuring point as:(2)$${\left.\tau (t)\right|}_{r,\varphi ,\psi =\mathrm{const}}\approx \sum_{m=0}^{n}\left(\;\alpha_{m}+\beta_{m}\frac{t}{T_{0}}\;\right)e^{-mt/T_{0}}\;\;\Rightarrow \;\;\dot{\tau }\left(t\right)=\frac{1}{T_{0}}\sum_{m=0}^{n}\left[\beta_{m}\;\left(\;1\;-\;\frac{mt}{T_{0}}\;\right)-m\alpha_{m}\right]e^{-mt/T_{0}},$$
where *n* and $T_{0}$ are constants subject to prior choice based on the physical behavior of thermal processes in the particular type of inertial sensor where the temperature is being measured, and $\alpha_{m}$, $\beta_{m}$ are to be determined further, assuming them to be constant locally. Depending on the purpose of approximation, we may take a different number of terms in (Equation 2) later.

As for the time derivative of the temperature, at this point we can compare one given by an ordinary averaging numerical differentiator with, e.g., a 3 min moving window, against that obtained from an analytic differentiation of the exponential approximation (Equation 2), both shown in Figure 3. What we call the ordinary averaging numerical differentiator here implies just a mean value of the first-order divided differences over a fixed window, according to the formula below:(3)$$\hat{\dot{\tau }}\left(t\right)=\frac{1}{N}\times \;\;\;\;\;\sum_{t-T<t_{i}⩽t}\;\;\;\;\;\frac{\tau^{\prime }\left(t_{i}\right)-\tau^{\prime }\left(t_{i-1}\right)}{t_{i}-t_{i-1}}\approx \frac{\tau^{\prime }\left(t\right)-\tau^{\prime }\left(t-T\right)}{T},\;T=3\;\min,$$
where $\hat{\dot{\tau }}$ stands for the temperature time derivative obtained using the ordinary numerical differentiator, and *N* denotes the number of temperature measurements $\tau^{\prime }\left(t\right)$ falling within a pre-defined window of length *T*. On a uniform time grid the average becomes even simpler, i.e., the first difference over the whole window.

In our example, the original data sampling frequency is 400Hz, and thus $N=7.2\;\times \;10^{4}$ given T=3min. Please note that when averaging over such a big number of short time steps, the order of difference taken in (Equation 3) virtually does not matter since the additional terms in them only generate some tiny edge effects. This estimate and the analytic one tend to agree with each other, but it is clear that plain averaging even over as little as 3 min in (Equation 3) introduces visible phase delays (see inset in Figure 3), and still suffers from quantization, which introduces errors, often up to 1 °C/hr. Note that the global exponential approximation here does not account for local short-term temperature variations, and therefore should not be taken as a true reference. However, instant jumps in the estimated value by 1 °C/hr, and sometimes by even 2 °C/hr, which the numerical differentiator inherits, definitely imply errors of that magnitude, given that temperature change in an IMU is a smooth and relatively slow process.

General theory dictates that with the time constant of the digital filter increased, phase delays will grow even bigger, though decreasing quantization error, and vice versa. From our experience, in some cases the balance between the two seems to be impossible to work out.

Thus, we draw three general conclusions from the considerations above:Exponentially decaying functions are good for describing thermal processes in inertial navigation units of navigation grade;In post-processing, we have a means of obtaining a rather accurate reference for the time derivative of the measured temperature (which still may not account for local short-term temperature variations), by exponential approximation;Applying an ordinary moving-average numerical differentiator to the temperature sensor readings produces a derivative estimate far from the desired one.

### 2.2. Real-Time Estimation Issues

For the inertial system to produce navigation solution in real time, there exist several extra concerns to address in practical applications:The patterns of temperature variations inside the IMU change considerably with time, and therefore infinite-impulse-response filters should be avoided, as they tend to keep track of the whole measuring history;However, no considerably long temperature measuring history may be entirely stored in the IMU to feed into any conventional finite-impulse-response digital filter with its window long enough to catch slow thermal processes;Sudden jumps in the temperature time derivative estimate, when fed into the error compensation algorithm for inertial sensors, exert additional noise in their calibrated output. As integrated in further processing, it may then considerably bias the navigation solution over time. Therefore, such jumps are commonly believed to be undesirable;Quantization errors of measuring slow changing temperature appear to have no properties that are any close to those of Gaussian white noise.

In addition, as we have already stated before, the real-time navigation algorithm requires having an estimate for the temperature time derivative at high update rate. Hence, any method similar to our ordinary numerical differentiator does not fit the first three of the above requirements. We will address them by using a real-time Kalman filter modified to have a finite impulse response (FIR) for least-squares approximation of temperature variations by a function similar to that in (Equation 2). For the last item in the above list, the workaround may be as follows. It appears that in our case temperature sensor measurements contain new information only when the sensor output updates, i.e., switches from one quantization level to another, which happens comparatively rarely. At this moment, we may be quite sure that the true value is close to the middle between the two quantization levels, with an accuracy being usually much better than that of the whole bunch of preceding constant measurements in a row. Therefore, for the purpose of estimating the temperature time derivative, we will generally use temperature sensor output only when it changes, and take middle value as a measurement along with its time tag for updating. This dramatically reduces the number of measurements involved in the estimation process. In further formulas, we will denote these measurements as ${\tilde{\tau }}^{\prime }\left(t\right)$, so that:(4)$$\tau^{\prime }\left(t\;-\;\Delta t\right)\neq \tau^{\prime }\left(t\right)\;⟹\;{\tilde{\tau }}^{\prime }\left(t\right)=\frac{\tau^{\prime }\left(t\;-\;\Delta t\right)+\tau^{\prime }\left(t\right)}{2},$$
where Δt is the time step of whatever nominal frequency the inertial unit operates at.

### 2.3. Reference Approximation

In order to evaluate the performance of the proposed technique, we need some kind of reference to compare against, which is the subject of this section. To demonstrate a more dynamic behavior of the temperature time derivative, we will now move to another data set with the results shown in Figure 4. This record holds temperature samples measured inside a quartz accelerometer, and includes variations in the temperature time derivative of roughly 1 °C/hr and more, up and down within approximately 10 min (see between minutes 60 and 70, 100 and 130, and 170 and 180 in the Figure). In this case, global approximation does not suffice the desired accuracy, and therefore we need a more improved method in order to obtain references for future comparison.

From the experience of processing multiple sets of data, the reference approximation described below both shows rather good accordance with common sense, and fits well the measured temperature in our records. Instead of making a global approximation as in Figure 2, we perform a moving fixed-length window local regression, considering Model (Equation 2). At this stage we are able to vary approximation parameters until the result becomes acceptable to serve as a reference. For this data, we chose a 4 min window, and fit function of the two terms from (Equation 2):$$\tau \left(t\right)\approx \tilde{\tau }\left(t\right)=\left[\alpha_{0}\left(t\right)+\beta_{0}\left(t\right)\frac{t}{T_{0}}\right]+\left[\alpha_{1}\left(t\right)+\beta_{1}\left(t\right)\frac{t}{T_{0}}\right]e^{-t/T_{0}},$$
with time constant $T_{0}$ of 30 min, and $\alpha_{0}$, $\beta_{0}$, $\alpha_{1}$, and $\beta_{1}$ chosen from a least-squares optimization over a window of length T=4min. Applied locally, this approximation differs from the original sensor measurements by less than the quantization step for the whole record, and provides a reasonable sense of time derivative, albeit ignoring some of its shortest-term variations on the time scale of 1–2 min and below. The latter requires even more thorough fine-tuning of reference approximation for each particular record, which we decided to omit as negligible. For most epochs, the window is symmetric to mitigate phase shifts introduced by the approximation. For all t<T/2 we use a fixed optimization window [0,T], and similarly for the epochs at the end of the record. This reduces edge effects to the most achievable extent.

Least-squares optimization also provides a covariance matrix to compute the standard deviation of estimated temperature and its derivative due to measurement noise, as follows. Consider estimating $\alpha_{j}$, $\beta_{j}$ over the centered window by least-square fit using all *k* measurements ${\tilde{\tau }}^{\prime }\left(t_{j}\right)$ described in Section 2.2 by (Equation 4), whose time instants $t_{j}$ fall within the approximation window:(5)$$\left[\begin{array}{c} \alpha_{0}\left(t\right)\\ \beta_{0}\left(t\right)\\ \alpha_{1}\left(t\right)\\ \beta_{1}\left(t\right) \end{array}\right]={\left(H^{T}H\right)}^{-1}H^{T}\left[\begin{array}{c} {\tilde{\tau }}^{\prime }\left(t_{1}\right)\\ \vdots \\ {\tilde{\tau }}^{\prime }\left(t_{k}\right) \end{array}\right]\;,\;H=\left[\begin{array}{cccc} 1 & \frac{t_{1}}{T_{0}} & e^{-\frac{t_{1}}{T_{0}}} & \frac{t_{1}}{T_{0}}e^{-\frac{t_{1}}{T_{0}}}\\ \; & \; & \vdots \\ 1 & \frac{t_{k}}{T_{0}} & e^{-\frac{t_{k}}{T_{0}}} & \frac{t_{k}}{T_{0}}e^{-\frac{t_{k}}{T_{0}}} \end{array}\right]\;.$$

In this notation, the covariance matrix for those coefficients appears to be
(6)$$P={\left(H^{T}H\right)}^{-1}\sigma_{0},$$
with $\sigma_{0}$ standing for standard deviation of a single measurement ${\tilde{\tau }}^{\prime }\left(t_{j}\right)$, which we may accept to be $\delta_{\tau }/4$, i.e., the quarter of a quantization step $\delta_{\tau }$. Remember that we are not taking the temperature sensor output itself, but the half-sum of the two with a strong negative correlation between them, as happens when the sensor reading updates from one quantization level to another with the true temperature being somewhere in between and close to the middle. From those covariances of coefficient estimates we then derive the estimated standard deviation $\sigma [\dot{\tilde{\tau }}]$ for temperature time derivative $\dot{\tilde{\tau }}$, determined by those coefficients:(7)$$\dot{\tilde{\tau }}\left(t\right)=\frac{\partial h}{\partial t}\left[\begin{array}{c} \alpha_{0}\left(t\right)\\ \beta_{0}\left(t\right)\\ \alpha_{1}\left(t\right)\\ \beta_{1}\left(t\right) \end{array}\right]\;,\;\frac{\partial h}{\partial t}=\left[\begin{array}{cccc} 0 & \frac{1}{T_{0}} & -\frac{1}{T_{0}}e^{-\frac{t}{T_{0}}} & \left(1\;-\;\frac{t}{T_{0}}\right)e^{-\frac{t}{T_{0}}} \end{array}\right]\;⟹\;\sigma \left[\dot{\tilde{\tau }}\right]=\sqrt{\frac{\partial h}{\partial t}P\frac{\partial h^{T}}{\partial t}}.$$

The 2-σ (95%) confidence interval is then displayed in orange beside the estimate plot in Figure 4. It would be unsound not to mention here, however, that the above assessments of estimation error magnitude account only for measurement errors, yet not for other kinds of errors, such as the discrepancy between the proposed model and true thermal processes, phase delay (also known as latency), etc. This remark completes the description of approximations accepted as references in this work.

### 2.4. Recurrent Least Squares Using Kalman Filter Update

As real-time implementation is the main subject to be established in this paper, we ought to describe briefly how the least-squares fit is being performed in its recurrent form. Basically, it is practically equivalent to the Kalman estimation problem [3] with the state vector *x* containing the desired coefficients of Model (Equation 2), assumed to be constants within the approximation time segment, so that the prediction equation becomes trivial:(8)$$\dot{x}=0.$$

We add also an initial estimate $\tilde{x}\left(0\right)$, taken zero, with its corresponding error covariance matrix P(0). Since our initial guess is an arbitrary one, P(0) is a diagonal matrix with some large numbers treated as numerical infinity in its diagonal. Thus, we established a base for our recurrent procedure.

At each time instant $t_{j}$, when a new measurement *z* is detected (see Section 2.2), we should update our estimate of the state vector *x* according to this new information, which is related to the state vector through a row matrix $h(t_{j})$:$$z\left(t_{j}\right)=h\left(t_{j}\right)x+r\left(t_{j}\right),$$
where *r* represents measurement error. All errors are assumed to be independent of each other and to have the same a priori standard deviation of $\sigma_{0}$. Let us now describe the step of recursion, which assumes that an a priori estimate ${\tilde{x}}^{-}$ is available along with its covariance matrix $P^{-}$, obtained from some *k* measurements equivalently to (Equation 5). The new measurement then becomes the (k+1)-th one, and we aim to derive an updated estimate ${\tilde{x}}^{+}$ and its covariance matrix $P^{+}$.

The Sherman–Morrison–Woodbury formula [6] for a single-rank modification of an inverted matrix, as applied to (Equation 5) and (Equation 6), respectively, gives:(9)$${\tilde{x}}^{+}={\tilde{x}}^{-}+\frac{P^{-}h^{T}\left(z-h{\tilde{x}}^{-}\right)}{\sigma_{0}^{2}+hP^{-}h^{T}},\;P^{+}=P^{-}-\frac{P^{-}h^{T}hP^{-}}{\sigma_{0}^{2}+hP^{-}h^{T}}.$$

However, from our experience, the above formula for the covariance matrix should never be used in practical implementations. Since it does not guarantee positive definiteness, given enough steps of recursion, numerical errors will eventually cause the matrix to fail to represent error covariance. To overcome this, one ought to use numerically stable square-root implementation, and we will stick here to the Cholesky upper-triangular square root form [7], such that
(10)$$P=SS^{T},\;S_{ij}=0\;\forall i>j,\;i.e.\;S=\mathrm{chol}\left(P\right).$$

Appendix A provides the corresponding procedure for the decomposition (Equation 10). Having this, the second equation in (Equation 9) becomes:(11)$$S^{+}=S^{-}\mathrm{chol}\left(E-\frac{S^{-T}h^{T}hS^{-}}{\sigma_{0}^{2}+hS^{-}S^{-T}h^{T}}\right)\;,$$
with *E* standing for the corresponding identity matrix. A detailed algorithm for the update step with Cholesky factorization implicitly embedded may be found in Appendix C, referred to as kalman_update further on.

### 2.5. Cancelling Prior Updates

To ensure finite impulse response of the approximation, we need an algorithm that cancels particular updates which were based on older measurements. This operation is just the reverse of those in (Equation 9) and (Equation 11):(12)$${\tilde{x}}^{-}={\tilde{x}}^{+}-\frac{P^{+}h^{T}\left(z-h{\tilde{x}}^{+}\right)}{\sigma_{0}^{2}-hP^{+}h^{T}},\;S^{-}=S^{+}\mathrm{chol}\left(E+\frac{S^{+T}h^{T}hS^{+}}{\sigma_{0}^{2}-hS^{+}S^{+T}h^{T}}\right)\;.$$

Note that Equations (Equation 12) only apply when the remaining measurements (after cancelling the former update) contain enough information for the state vector estimate to stay well-conditioned, i.e., uniquely determined by them. This ensures all operations on the right side to make sense. Please refer to Appendix D for component-wise implementation of the procedure, which is very similar to the previous algorithm, only having opposite signs in several places. We will later refer to this implementation as kalman_pullback.

## 3. Real-Time Filtering

Having all background models and procedures now described, we may proceed with the real-time algorithm for estimating the temperature time derivative. Firstly, we modify the model function (Equation 2) so as to prevent numerical errors when the exponent vanishes compared to other terms for large *t*. Secondly, we choose a minimum number of terms in our model, to make it the sum of three functions, which seem to be enough for approximating the temperature within several minutes. Altogether, we have:$$\tau \left(t\right)\approx \tilde{\tau }\left(t\right)=\alpha +\beta \frac{t-t_{0}}{T_{0}}+\gamma e^{-\left(t-t_{0}\right)/T_{0}},$$

Which is next represented as:(13)$$\tilde{\tau }\left(t\right)=h\left(t\right)x,\;h\left(t\right)=\left[\begin{array}{ccc} 1 & \frac{t-t_{0}}{T_{0}} & e^{-\left(t-t_{0}\right)/T_{0}} \end{array}\right]\;,\;x=\left[\begin{array}{c} \alpha \\ \beta \\ \gamma \end{array}\right]\;⟹\;\dot{\tilde{\tau }}=\frac{\partial h}{\partial t}x,$$
where the parameters α, β, γ, $t_{0}$, and $T_{0}$ are assumed constant locally, but may change over the course of the system operation. Among them, α, β, and γ are subject to estimation using recurrent least-squares FIR approximation. The time bias $t_{0}$ serves to prevent the exponent from becoming too small compared to other coefficients. The time constant $T_{0}$ may be the subject of estimation using nonlinear filtering, which is reserved for future study. Our current research shows that with $T_{0}$ taken constant within a reasonable range the filter works well too, regardless of the particular value of $T_{0}$.

The length $T_{w}$ of the approximation window is another issue to handle. On the one hand, it should not be too small for the approximation to average out measurement errors sufficiently. On the other hand, when fixed to be too large, it may suppress short-term thermal variations. Therefore, we fix its minimum value, allowing the approximation window to expand if too few measurements fall within it. The latter means that the temperature changes so slowly, that extending the approximation window seems to be rather natural.

With the above considerations put forward, the real-time algorithm basically consists of five principal parts, with their flowchart shown in Figure 5:Initialization;Detecting the new measurements and applying them, if any;Checking for the expired measurements and retracting them from estimates, if found;Amending the model time bias $t_{0}$, if needed;Updating the current estimate for temperature and its time derivative;

We will recount these main pieces of real-time processing and their integration below. In total, the implementation contains roughly under a hundred lines of code, which makes up less than one per cent of a typical software project in a real inertial navigation system. Note that all measurements that are currently used in the estimation process are to be stored in a buffer along with their time tag, until they are considered as expired and pulled back from the estimation. More specific definitions follow.

### 3.1. Initialization

Even though, strictly speaking, only a few measurements are enough to determine Model (Equation 13), and the values that give a start to the recurrent process therefore may seem to be irrelevant, two factors still require special attention. Both of them are to be understood through a more detailed examination of the initial stage of reading temperature sensors in the example of Figure 1. It illustrates the possibility of receiving constant values for approximately 5 min in a row. We should recall once more here that only measurements described by (Equation 4) contain new information about temperature variation. Therefore, such measurements may simply not appear initially for a time. The second likely problem is that even if the measurements do appear within the first minute of operation, as in Figure 1, those few may lie so close to each other in time, that the derived model coefficients will be ill-conditioned. Therefore, more consideration should be given to the starting procedure.

In order to mitigate the transition process, for n=3 estimated model coefficients we recommend generating 2n=6 “fake” constant measurements lying in the negative time domain and evenly separated by 1 min from each other. The constant taken is to be the first valid reading from the temperature sensor $\tau^{\prime }\left(0\right)$, so that these virtual measurements become
(14)$${\tilde{\tau }}^{\prime }\left(t_{k}\right)\equiv \tau_{0}=\tau^{\prime }\left(0\right),\;k=-2n,\cdots ,-1,\;t_{k}=k\;\mathrm{minutes}.$$

We then set $t_{0}$ to −2n, initial estimate $\tilde{x}$ to all zeros, and starting covariances to some big numbers (e.g., $10^{6}$ °C), and go through 2n steps of the Kalman update procedure from Section 2.4 applied to (Equation 13) and (Equation 14). Having done this, we obtain a starting estimate $\tilde{x}\left(0\right)$ and its corresponding covariance upper-triangular square root S(0). This initialization implies having a zero derivative at t=0 (which makes sense on system startup), and even if its true value is far from that (e.g., on restart), real measurements will soon follow and override those virtual ones very quickly.

### 3.2. Updating Coefficient Estimates

According to (Equation 4), each time the temperature sensor output switches between two quantization levels, we handle this event as an update. Additionally, an update is required if temperature prediction based on the model deviates too much from the actual sensor reading. We may suggest treating the current model as unfit when the deviation exceeds the quantization step. In this case, the temperature sensor reading itself serves as measurement ${\tilde{\tau }}^{\prime }$:(15)$$\left|h\left(t\right)x-\tau^{\prime }\left(t\right)\right|>\delta_{\tau }\;⟹\;t_{k}=t,\;{\tilde{\tau }}^{\prime }\left(t_{k}\right)=\tau^{\prime }\left(t\right).$$

Either coming from (Equation 4) or (Equation 15), both new measurement ${\tilde{\tau }}^{\prime }\left(t_{k}\right)$ and its time tag $t_{k}$ are stored in a measurement buffer. If sensor readings contain an apparent noise, so that these update events happen too often to store all of them in a buffer of a limited size, an appropriate averaging is recommended. In this case, we should note, a conventional numerical differentiation and the filter described here may perform very similarly, and there might be no particular preference of one of these methods over another.

Once detected at $t=t_{k}$, the measurement (Equation 4) or (Equation 15) goes into the Kalman update procedure described in Section 2.4 with $h=h(t_{k})$, state vector *x* according to (Equation 13), and an a priori standard deviation for the measurement error $\sigma_{0}=\delta_{\tau }/4$ (for $\delta_{\tau }$ being the quantization step). The procedure yields an updated estimate $\tilde{x}$ for the state vector and its upper-triangular square root covariance matrix *S*. Hereby, coefficient estimates lying inside the state vector *x* are being updated.

### 3.3. Retracting Expired Measurements

We may consider two ways of preventing the filter from keeping track of an improperly long measurement history. The first one is inherent to the standard Kalman filtering procedure, implying the introduction of non-zero noise into the prediction equation (Equation 8), and its corresponding covariances into the prediction step of the Kalman filtering procedure for covariance [3]. However, in our case the accepted model (Equation 13) has a mostly empirical nature, and therefore we find it difficult to establish any reasonable quantitative connection between the supposed changes in thermal processes and in the model coefficients over time. Simply put, we have not yet discovered a way to determine the above noise covariances that would produce any reasonable result whatsoever. Thus, we adopt complete ridding of the measuring history from some prior point in time by making our filter have a finite impulse response. This is performed through what we call here the retraction of expired measurements, as follows.

Given an a priori chosen minimum approximation window length $T_{w}$, we consider a particular measurement ${\tilde{\tau }}^{\prime }\left(t_{k}\right)$ as an expired one, under three conditions:It falls behind the current time *t* by more than $T_{w}$: $t-t_{k}>T_{w}$;The measurement next to the one under consideration, i.e., ${\tilde{\tau }}^{\prime }\left(t_{k+1}\right)$, stays behind the newest measurement used for updating (see Section 3.2) by $T_{w}$ or more: $t_{l}-t_{k}⩾T_{w}$, with $t_{l}$ standing for the time tag of the newest measurement;A number of measurements of at least twice the state vector component count will remain used for estimation: l−k+1>2n.

Conditions 2 and 3 ensure that, after removing ${\tilde{\tau }}^{\prime }\left(t_{k}\right)$, the estimate for the state vector still stays well-conditioned. If all three are satisfied, the measurement ${\tilde{\tau }}^{\prime }\left(t_{k}\right)$ is subject to a retraction procedure described in Section 2.5, with $h=h(t_{k})$ according to (Equation 13). It replaces the estimated $\tilde{x}$ for the state vector and its upper-triangular square root covariance matrix *S* with new ones, as well as the coefficients in (Equation 13). In a practical implementation, one more option to treat the oldest measurement in the buffer as expired emerges when the buffer overflow is due for the next epoch.

### 3.4. Amending Model Time Bias

Once all expired measurements have been removed, we amend the model time bias $t_{0}$ in (Equation 13). Without performing this, given enough time of operation, the coefficients in the model will eventually become improperly unbalanced in magnitude, introducing substantial numerical errors.

Let ${\tilde{\tau }}^{\prime }\left(t_{o}\right)$ be the oldest measurement remaining into estimation after retracting all expired measurments out of it. We now replace $t_{0}$ in (Equation 13) by $t_{o}$ using the following formulas:(16)$$C=\left[\begin{array}{ccc} 1 & \frac{t_{o}-t_{0}}{T_{0}} & 0\\ 0 & 1 & 0\\ 0 & 0 & e^{-\left(t_{o}-t_{0}\right)/T_{0}} \end{array}\right]\;,\;{\left.\tilde{x}\right|}_{t_{o}}=C{\left.\tilde{x}\right|}_{t_{0}},\;{\left.S\right|}_{t_{o}}=\mathrm{chol}\left(C{\left.S\right|}_{t_{0}}{\left.S\right|}_{t_{0}}^{T}C^{T}\right)\;.$$

After that, $t_{0}$ is replaced by $t_{o}$ from this point on, until the next amendment emerges. We will refer to this piece of processing as amend_t0 as described in Appendix B.

### 3.5. Updating Temperature and its Time Derivative

This calculation is performed at each time epoch, independent of whether any measurements have been updated or have expired. For the current estimated temperature and its rate of change, we use (Equation 13) with *x* taken as its current estimate from the above algorithm. For the estimated standard deviation σ[·] of their errors, as in (Equation 7), the following expressions hold:(17)$$\sigma \left[\tilde{\tau }\left(t\right)\right]=\sqrt{h\left(t\right)SS^{T}h{\left(t\right)}^{T}},\;\sigma \left[\dot{\tilde{\tau }}\left(t\right)\right]=\sqrt{\frac{\partial h(t)}{\partial t}SS^{T}\frac{\partial h{\left(t\right)}^{T}}{\partial t}}.$$

Again, the estimates in (Equation 17) account only for measurement errors, and not for those from other sources (see the last paragraph of Section 2.3). However, these quantities do serve as good indicators of observability and convergence.

### 3.6. Real-Time Filtering Algorithm at a Glance

In this section, we provide the whole algorithm according to the flowchart in Figure 5. Its pieces are supposed to reside within a certain real-time on-board navigation processing, which provides the current temperature sensor reading $\tau^{\prime }$ and its time tag *t*. Estimates for the temperature and its time derivative, $\tilde{\tau }$ and $\dot{\tilde{\tau }}$, respectively, serve as the algorithm output to be used further in thermal compensation of inertial sensor errors.

Let us arrange the algorithm below. Table 1 and  Table 2 describe input and output quantities, and Listings 1 and 2 contain operations to be put into the initialization and the main cycle, respectively, of the IMU algorithm. Double slashes (//) separate comments in a C-like manner. All time tags and constants are supposed to be measured in seconds.

**Listing 1.** Initialization part.

define x = n×1 zero matrix // current estimates for model coefficients

define S = n×n identity matrix // upper-triangular square root of covariance

define Z = N×2 zero matrix // measurement buffer, 2-nd column for time tags

σ0 = δτ/4 // a priori standard deviation of single measurementτ0 = τ′ // initial temperature sensor readingS = 10^6^·S // initial covariance square rootfor i = 1..2·n increasing // go through fake initial measurements      Zi1 = τ0 // store measurement value in buffer      Zi2 = t − (2·n+1−i)·60[seconds] // store time tag      (x, S) = kalman_update(x, S, Zi1, h(Zi2), σ0, n) // update estimatesend for ii0 = 1 // oldest measurement index in bufferi1 = 2·n // newest measurement index in buffert0 = Z12 // model time bias



**Listing 2.** Main cycle portion

... // IMU sensor acquisitionτ1 = τ′ // next reading from temperature sensorif |τ1−τ0|>δτ/2 | |τ˜−τ0|>δτ // temperature sensor has updated, or model deviates  i1 = i1 + 1  if i1>N then i1 = 1 end // buffer rollover  Zi11 = (τ1+τ0)/2 // store middle value as measurement  Zi12 = t // store time tag  (x, S) = kalman_update(x, S, Zi11, h(Zi12), σ0, n) // update estimatesend if |τ1−τ0|
j
${}_{0}$
 = i
${}_{0}$
 // store oldest measurement index, check for expired or buffer overrun

while ((t−Z
${}_{i_{0}2}$
>T
${}_{w}$
) & (Z
${}_{i_{1}2}$
−Z
${}_{i_{0}2}⩾$
T
${}_{w}$
) & (mod(i
${}_{1}$
−i
${}_{0}$
,N)⩾2·n)) | mod(i
${}_{0}$
−i
${}_{1}$
,N)=1

  (x, S) = kalman_pullback(x, S, Z
${}_{i_{0}1}$
, h(Z
${}_{i_{0}2}$
), 
$\sigma_{0}$
, n) // retract oldest one

  i
${}_{0}$
 = i
${}_{0}$
 + 1

  if i
${}_{0}$
>N then i
${}_{0}$
 = 1 end // buffer rollover

end while

if j
${}_{0}\neq$
i
${}_{0}$
 // oldest measurement changed

  (x, S) = amend_t0(Z
${}_{i_{0}2}$
, Z
${}_{j_{0}2}$
, x, S, T
${}_{0}$
) // amend t
${}_{0}$
 in estimates

  t
${}_{0}$
 = Z
${}_{i_{0}2}$
 // new model time bias

end if j
${}_{0}$


$\tilde{\tau }$
 = h(t)·x // temperature estimate

$\dot{\tilde{\tau }}$
 = 
$\frac{\partial h(t)}{\partial t}\cdot$
x // time derivative estimate

$\sigma_{\tau }$
 = 
$\sqrt{\;h(t)\cdot S\cdot S^{T}\cdot {h(t)}^{T}\;}$
 // temperature standard deviation estimate

$\sigma_{d}$
 = 
$\sqrt{\;\frac{\partial h(t)}{\partial t}\cdot S\cdot S^{T}\cdot {\frac{\partial h(t)}{\partial t}}^{T}\;}$
 // time derivative standard deviation estimate

$\tau_{0}$
 = 
$\tau_{1}$
 // store previous measurement

... // proceed with IMU inertial sensor error compensation & navigation




## 4. Results

In this section we provide illustrations for the described real-time filter applied to sample data sets. To give a general impression of how the estimate may look like, Figure 6 displays a temperature time derivative obtained by our filter for the same data sets as in Figure 2 and Figure 4, respectively, as compared to those of a reference exponential approximation and of an ordinary averaging numerical differentiator. For the first data set, the real-time filter window has a length $T_{w}$ of 5 min, and the model time constant $T_{0}$ measures 3 min. Since the second sample comes from the more thermally dynamic interior of an accelerometer, where the heating rate reaches its peak twice faster than in the record from a ring laser gyroscope (as seen from the Figure), the above values were appropriately adjusted to be 3 and 2 min, respectively. Their temperature time derivative estimates overlay the reference within a tolerable margin, having an apparent few-minute phase delay when the slope changes too fast, which seems to be unavoidable to a certain extent. It is believed though, that thermal processes in inertial measurement units are slow enough to accept a latency of 1–2 min in measuring the heating rate.

More detailed comparative analysis follows in the next sections.

### 4.1. Real-Time FIR Filter Compared to Conventional Kalman Filtering

A conventional Kalman filter, given its finite values of noise covariances, essentially has an infinite impulse response. Nevertheless, one may employ the option of adjusting those variances appropriately in order to get the desired properties of the filter. This often works perfectly. However, in our case the empirical model that we use has not been analytically derived from particular physical properties of the entity inside of which the temperature is being measured. Therefore, no reasonable assumptions may be laid on its coefficients in terms of their dynamics over time.

In spite of the above, for just three coefficients of our model, a virtually exhaustive discrete numerical search may be performed to discover whether there exists such a combination of variances that allows producing an acceptable result by conventional Kalman filtering. And it has appeared that there seems to be none. Figure 7 features the most adequate results obtained so far, which still look nowhere near the desired approximation. With the parameters used to obtain the results shown, the filter generates an estimate that generally follows the true value, albeit being quite far from it. The filter also provides a rather adequate estimate for its 2-σ deviation, which has a magnitude of several degrees per hour. From this, we conclude that conventional Kalman filtering, at least with the same simple model, is barely able to outperform the filter that has a truly finite impulse response while solving this estimation problem.

### 4.2. Using a Plain Linear Model Function

The next point to examine lies in the opportunity of using an even simpler model than that of (Equation 13). In particular, we have tested if the exponential term there may be omitted, leaving a bare linear function with only two coefficients to estimate, and no need to handle the model time bias $t_{0}$. This, indeed, appears to be possible to some extent.

For short approximation windows, and after heating becomes slow enough, the results seem to look indiscernible. However, two major drawbacks of the plain linear function emerge in the beginning, as displayed in Figure 8 for the approximation window of length $T_{w}=5\;min$:The linear function tends to over-estimate its own prediction accuracy;The linear function introduces larger phase delays (i.e., latency of the estimate).

Therefore, the exponential model function (Equation 13) is still preferable for having less latency and a more appropriate accuracy estimation.

### 4.3. Modifying Time Parameters

As for the time parameters of the real-time filtering procedure, the minimum approximation window length $T_{w}$ and the time constant $T_{0}$ are subjects to adjustment when transferring the method between different systems. The reasoning behind their selection is as follows. Firstly, $T_{0}$ should be several times less or equal to $T_{w}$, in order for both coefficients of linear and exponential terms in Model (Equation 13) to be well observable. Future study may reveal whether $T_{0}$ should join a state vector, thus forcing the filter to become a nonlinear one.

Secondly, one ought to be conscious that short-term temperature variations may get lost within roughly a half of the minimum window length $T_{w}$. Furthermore, an average latency of that magnitude is generally expected for the estimates produced by the filter. However, thermal processes in inertial units of navigation grade tend to be slow enough to accept values starting from several minutes. As for the maximum value, we may suggest choosing it from the range of 3–10 min after gaining some experience of processing measurements from the particular type of temperature sensor. In our case, it is quite definite that thermal processes in a ring laser gyro, with its sensing elements operating in low-pressure gas inside of a glass housing, are not as rapid as those of a much smaller accelerometer made mostly of metal. We therefore may take the minimum approximation window length for a sensor inside the gyroscope as larger as roughly twice of that for the accelerometer. One should also keep in mind the general relation between this parameter, the quantization step $\delta_{\tau }$ (which occurred to be 0.05 °C, 0.1 °C, and 0.2 °C for different systems in our experience), and the expected estimation accuracy $\Delta \dot{\tilde{\tau }}$:$$\Delta \dot{\tilde{\tau }}∼\delta_{\tau }/T_{w}.$$

In our examples we have $\Delta \dot{\tilde{\tau }}∼$ 0.05 °C/3 min = 1 °C/hr, which seem to conform with what we see and with what we aim at. However, the difference between the two estimates obtained using different values of $T_{w}$ does not look too striking, as Figure 9 suggests. We treat both estimates as satisfactory, and therefore both values of $T_{w}=3\;\min$ and $T_{w}=5\;\min$ as acceptable, too. Still, a longer window obviously produces larger latencies.

As for the time constant $T_{0}$, the sensitivity to its value is even lower. From Figure 10 we may observe that estimates are basically the same within small artifacts. However, the estimated standard deviation is considerably higher for larger $T_{0}$. Nevertheless, from some point in time this difference vanishes completely and never appears again.

## 5. Conclusions

In this work, we have addressed four primary concerns that may arise while estimating the temperature time derivative in an inertial measurement unit of navigation grade:Measurement quantization under slowly changing temperature;Wide time domain of thermal processes;No white noise properties in measurement errors;Real-time recurrent operation.

To overcome these issues, we have developed a finite-impulse-response recurrent filter based on a least-squares approximation of temperature measured only at quantized sensor output updates by an exponentially decaying function within an adaptive-length window, using the numerically stable square-root Kalman filtering. The method has performed well on hundreds of sample data sets recorded from a self-heating inertial measurement unit in a test environment.

To adapt the method to a specific system, one should pick a value of quantization step inherent to its temperature sensors, and decide on two time parameters as described in Section 4.3. If some items from the above list are not the case in a particular system, the method may be appropriately simplified.

Several comparative tests have been carried out. In-flight testing and improved non-linear filtering are suggested for future studies.

## Figures and Tables

**Figure 1 sensors-20-01299-f001:**
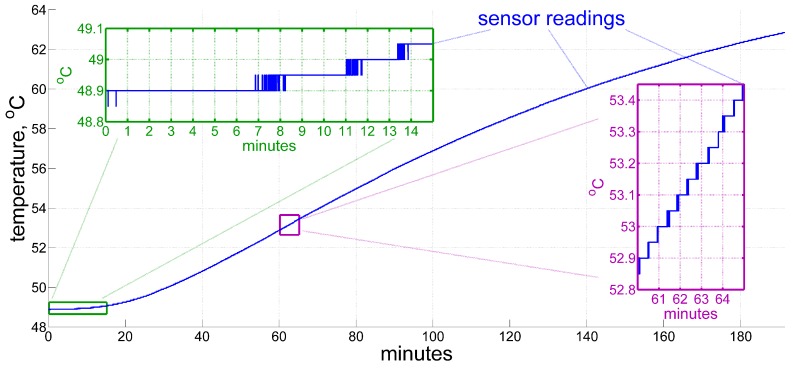
An example of a temperature sensor output under inertial measurement unit (IMU) self-heating. In the left inset the measurements stay constant for approximately 6–7 min, and then flicker for another minute due to quantization, although the temperature itself rises slowly during this period at an average pace of 0.5 °C/hr. By contrast, the heating rate appears to be of an order higher in the right inset, roughly 5 °C/hr. A real in-flight externally forced temperature variation may be even faster, up to 1–2 degrees per minute.

**Figure 2 sensors-20-01299-f002:**
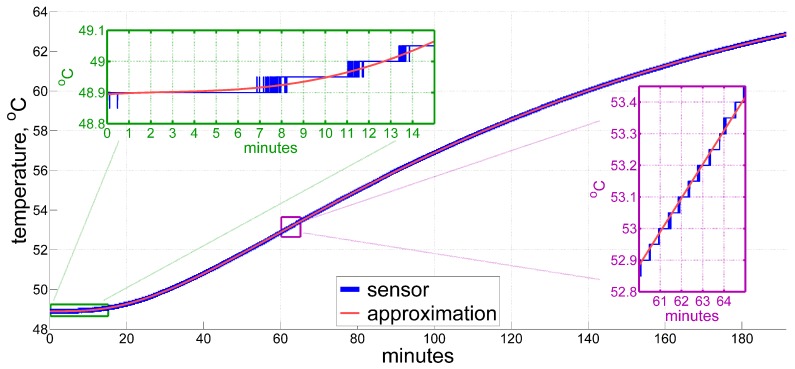
A global approximation of temperature using the exponential series: $\tau \approx 67.3+\sum_{m=1}^{6}\;\;d_{m}e^{-mt/90}$.

**Figure 3 sensors-20-01299-f003:**
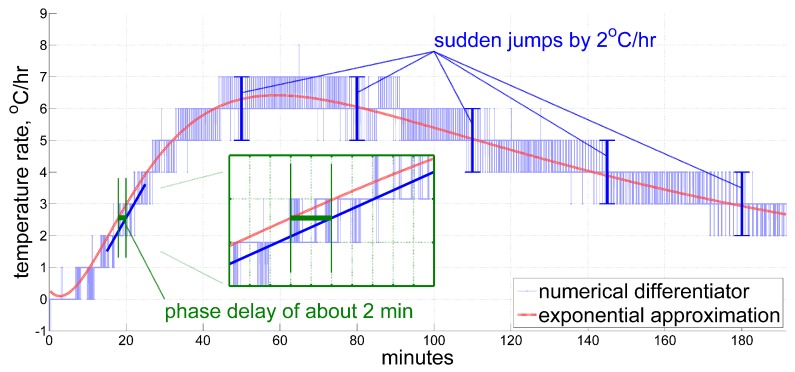
A time derivative of the exponential approximation compared against an ordinary averaging numerical differentiating filter with the latter one introducing phase delays and quantization errors.

**Figure 4 sensors-20-01299-f004:**
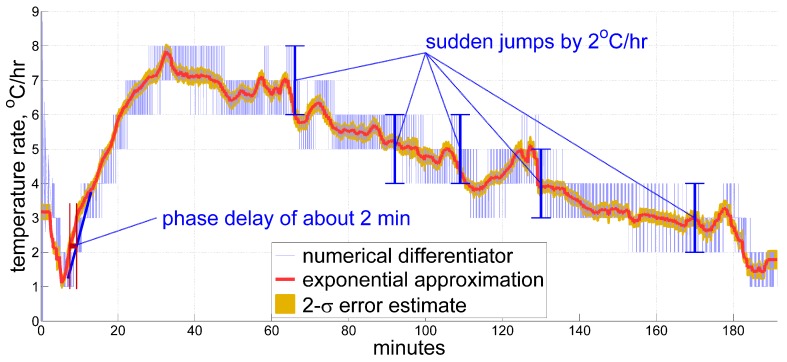
An example of a temperature time derivative estimate obtained from exponential approximation in post-processing to serve as reference. A record with stronger thermal variations under inertial unit self-heating shows fluctuations in the temperature rate by 1 °C/hr and above within 10-min and shorter time periods.

**Figure 5 sensors-20-01299-f005:**
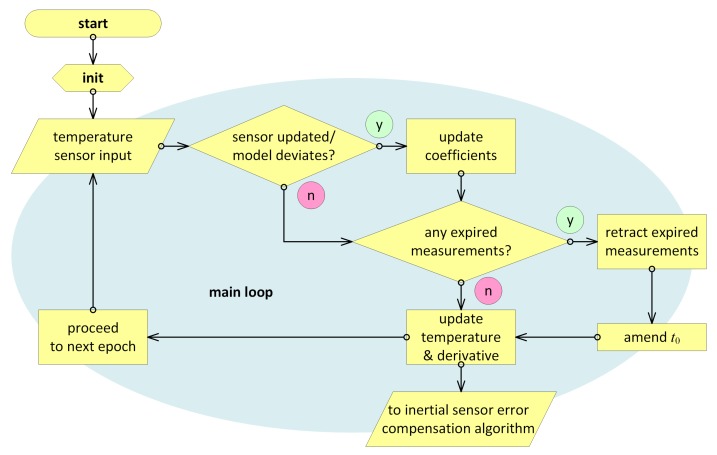
Real-time temperature time derivative estimation flowchart.

**Figure 6 sensors-20-01299-f006:**
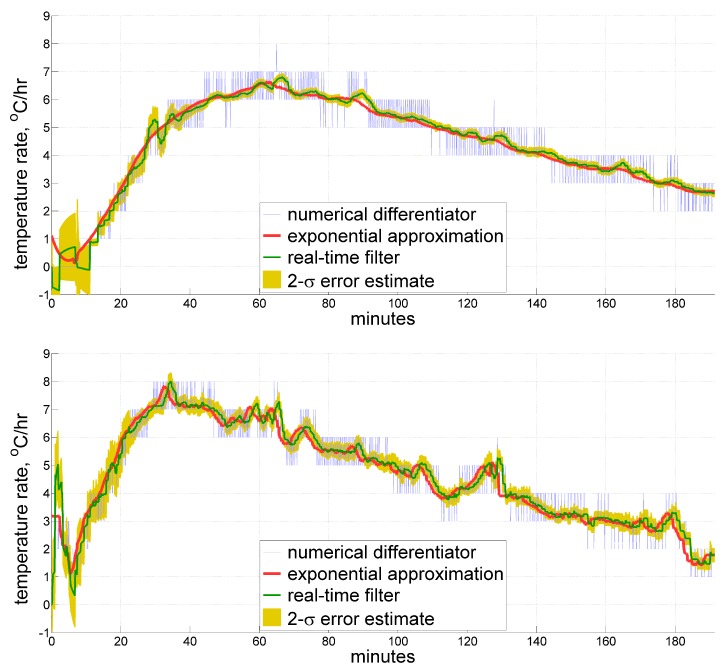
Estimates for the temperature time derivative obtained from thermal sensors inside a ring laser gyroscope (**top**) and an accelerometer (**bottom**) under IMU self-heating. Real-time filtering results are shown along with their 2-σ estimates, compared against the ordinary averaging numerical differentiator and reference exponential approximation, both derived in post-processing.

**Figure 7 sensors-20-01299-f007:**
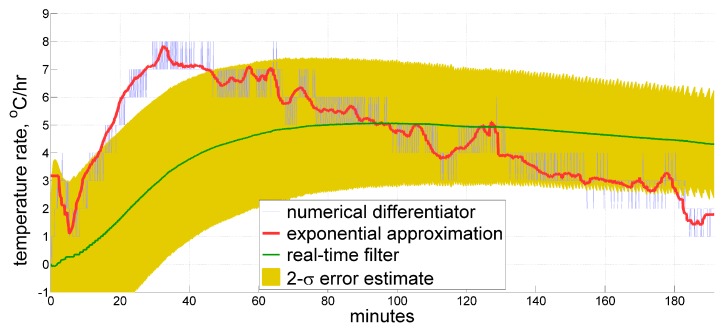
Estimate for the temperature time derivative produced by a conventional infinite-impulse-response real-time Kalman filter compared to the reference approximation.

**Figure 8 sensors-20-01299-f008:**
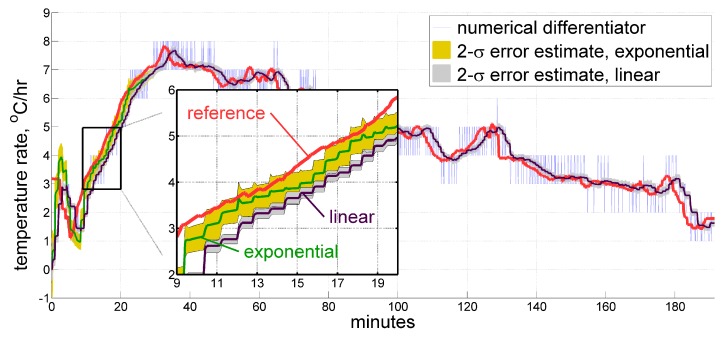
Estimate for the temperature time derivative produced by the real-time filter using exponential and linear model functions.

**Figure 9 sensors-20-01299-f009:**
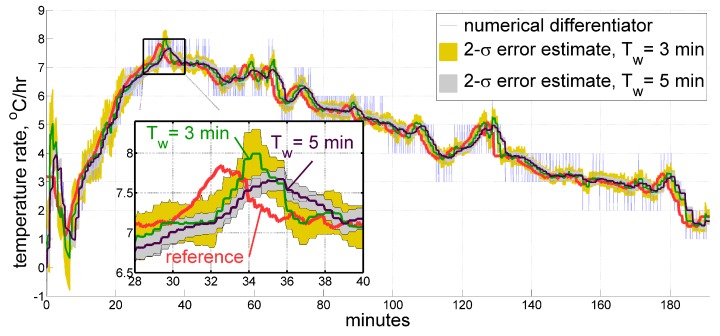
Estimate for the temperature time derivative produced by the real-time filter using different minimum approximation window lengths $T_{w}$ of 3 and 5 min over the same data set.

**Figure 10 sensors-20-01299-f010:**
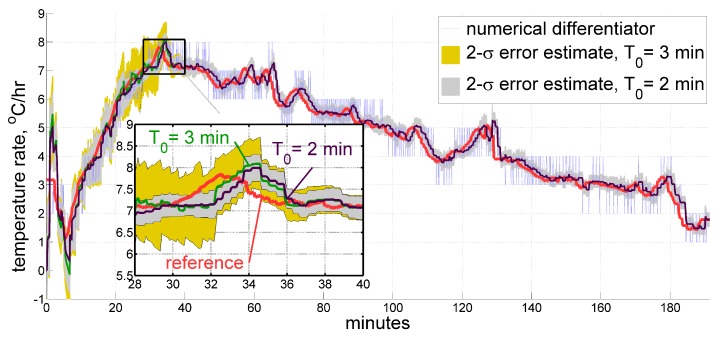
Estimate for the temperature time derivative produced by the real-time filter using different model time constants $T_{0}$ of 3 and 2 min over the same data set.

**Table 1 sensors-20-01299-t001:** Input quantities.

Variable	Meaning	Remarks
t	current time	at each epoch
$\tau^{\prime }$	current temperature sensor reading	at each epoch
$\delta_{\tau }$	quantization step	0.05 °C for sample data sets
$T_{w}$	minimum approximation window length	5 and 3 min for sample data sets
$T_{0}$	model time constant	3 and 2 min for sample data sets
n	model order	currently 3
N>2n	maximum number of measurements allowed to store in memory buffer	90 is enough for sample data sets, though multifold more recommended to reserve, if possible

**Table 2 sensors-20-01299-t002:** Output quantities.

Variable	Meaning	Remarks
$\tilde{\tau }$	current temperature estimate	at each epoch
$\dot{\tilde{\tau }}$	current temperature time derivative estimate	at each epoch
$\sigma_{\tau }$	estimate for temperature standard deviation	at each epoch
$\sigma_{d}$	estimate for temperature time derivative standard deviation	at each epoch

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
