# Peer review of "Real-Time Estimation of Temperature Time Derivative in Inertial Measurement Unit by Finite-Impulse-Response Exponential Regression on Updates"

_sensors, 2020, doi:10.3390/s20051299_

Round 1
Reviewer 1 Report
This paper present a study of modeling the sampled temperature measurements, with the aim of estimating the derivation of the temperature in real time. A new model, namely the exponetial model is employed to parameterize the time variation of the temperature, and based on this model, the temperature along with its derivative is estimated in real time.
This title cannot be said informative, the main approach employed in this study is completely missing in the title.
Author Response
Firstly, I would like to express my deep gratitude to the reviewer for spending their time reading through our manuscript.
As for the title of the paper, I will consult the Editor if it can be changed at this point of submission to become more informative. I strongly suppose it may become an issue, though.
The title revised now.
Reviewer 2 Report
The paper describes a numerical method for extracting precise temperature from slowly varying measured data.
The approach is sound and the paper looks free from major errors. Nevertheless, the method seems too complicated to be implemented in real systems. Moreover, the method is applied in a specific given scenario, being hard to export to different systems. In summary, in my opinion, the paper does not describe a real useful contribution.
Author Response
Firstly, I would like to express my deep gratitude to the reviewer for spending their time reading through our manuscript.
Regarding the comments, our response is as follows.
"the method seems too complicated to be implemented in real systems"
I fully recognize that rationale described in the text may seem overloaded for the purpose of rigor and background motivation for each statement made there. However, given the final algorithm listings provided in the paper, they all fit within half of the screen of source code or less, and work just fine in real time with no complicated fancy operations. Compared to real algorithms used in integrated navigation systems, with Kalman filtering of up to 30th order, and many many initializations, different operation regimes, and so on, our implementation is way too easier than most of processing parts in navigation solution.
"the method is applied in a specific given scenario, being hard to export to different systems"
The scenario which our method is being applied to is indeed not always the case. If it is not, then a lucky developer may be free of all the complications described in our paper. It happens, for example, when sensor noise exceeds the quantization step, and therefore natural dithering occurs. Although time derivative estimation out of measurements like these may seem simpler, it may become impossible in principle to assess its value accurately enough due to the very same noise, that generally degrades the measurement accuracy. In this case the problem just lies in another field. Nevertheless, if temperature sensors are precise enough, as in one of our systems, we may take an advantage of their precision by estimating temperature time derivative using our method.
For our algorithm, we have hundreds of records of thermal transition where it performs well enough to be called "tested and verified" between our engineers. It is also generalized to the extent of only a few tuning parameters, most of them rather straightforward, so we do not expect any trouble applying it in other systems. If you have temperature records that make you doubt it, you may send them to us so we could give a try.
"the paper does not describe a real useful contribution"
As I mentioned, the method described in our paper works well for all records we have (hundreds of them). Of course, there exist a problem of the same importance, which lies in establishing appropriate reliable models for inertial sensor errors produced by thermal effects in temperature transition. This is a contemporary developing field of study in inertial navigation, and our work is suggested to be a mere contribution to it, not a full final solution.
Reviewer 3 Report
In this paper, the authors reported a filtering technique that could be used for estimating the time derivative of slowly changing temperature measured via quantized sensor output in real-time. Also compare the technique against both ordinary averaging numerical differentiator and conventional Kalman filter, over a set of real samples recorded from the inertial units. The paper is well-organized and easy to follow. Only one small suggestion:
In all Figs, suggest using temperature label Co
Author Response
Firstly, I would like to express my deep gratitude to the reviewer for spending their time reading through our manuscript.
We changed all labels in our plots to look either like 'temperature, °C', or 'temperature rate, °C/hr'
Reviewer 4 Report
A filtering technique that allows estimating the time derivative of slowly changing temperature measured via quantized sensor output is developed. This technique relying on the physical nature of the internal thermal processes and it make it possible to minimize an error which corresponds to changing the temperature sensor reading by one single quantization step. A finite-impulse-response recurrent Kalman filter based on least-squares approximation of temperature measured has been developed which takes into account the exponentially decaying function within an adaptive-length window. The paper falls into the scope of the journal and has enough scientific merit.
It is recommended to shortening the sections 1. Introduction and 2.1. Thermal behavior. The section 4.2. Using plain linear model function can be omitted without detriment to the meaning of the paper.
Author Response
Firstly, I would like to express my deep gratitude to the reviewer for spending their time reading through our manuscript.
Regarding the comments, our response is as follows.
"It is recommended to shortening the sections 1. Introduction and 2.1. Thermal behavior"
Although these sections initially were, indeed, shorter, and some of their statements may now look trivial for a qualified reader, we added more explanations to them on request of our informal reviewers who have had less expertise in the field. From this perspective, we would prefer to keep those extended versions, since they suggest more background reasons for all assumptions and statements made there. We find it helpful to a general reader.
"The section 4.2. Using plain linear model function can be omitted without detriment to the meaning of the paper"
Comparing model function against the linear one seems to be essential part of the study to us, for the latter being equivalent to using standard differentiating filters of other kinds apart from the trivial one considered in the beginning of our paper. This comparison provides actual evidence of their drawbacks. We would therefore rather keep this part, should a reader doubt the role of the exponential term in our model function, regarding it as excessive.
Round 2
Reviewer 2 Report
Dear authors,
Thank you for your kind response.
Nevertheless, my opinion has not changed.
The paper describes a numerical method for extracting precise temperature from slowly varying measured data.
The approach is sound and the paper looks free from major errors. Nevertheless, the method seems too complicated to be implemented in real systems. Moreover, the method is applied in a specific given scenario, being hard to export to different systems. In summary, in my opinion, the paper does not describe a real useful contribution.
Author Response
"The paper describes a numerical method for extracting precise temperature from slowly varying measured data."
In fact, not temperature is the subject to estimation, but its time derivative, which makes the process more tricky under the conditions stated.
"the method seems too complicated to be implemented in real systems"
I strongly believe that less than 100 lines of source code should not be considered as too complicated for real system, whose software typically contains tens of thousands. This absolutely applies to any real system in industry. I have personal experience in writing source code for one of the mostly used inertial navigation system in our country (and for some others, as well, you can find my authorship in projects like these: ieeexplore.ieee.org/document/8769418, scopus.com/record/display.uri?eid=2-s2.0-84938887474&origin=inward&txGid=a053f1f68caa85373f4738e6458d7fc7, and ieeexplore.ieee.org/document/8943630). I have the source code right at work. It has more than 30.000 lines of code, so our algorithm is literally 0.3% of that. I hope this illustrates well that the method is indeed appropriate to be implemented in real system. I added it to Section 3.
However, if something really can be simplified and/or improved, I would kindly ask you to provide a clue on what and how. We have addressed all ways of simplifying our algorithm that seem evident to us, in Introduction, and Sections 2.2, 4.1, 4.2.
"the method is applied in a specific given scenario, being hard to export to different systems"
I am not sure if I can grasp the particular meaning of the term "scenario" here, but hundreds of records under different thermal variations has been processed so far. I believe that statement of method being hard to export to different systems should be supported by some reasoning behind it. Transferring anything from one system to another always takes some effort, and every method should be used appropriately when needed. In fact, for instance, sensor calibration and error compensation is much harder to export between different systems. However, it always becomes done eventually, and different methods are being used. And many much more complicated methods has to be implemented. So I am wondering why doubt that this rather simple one would not become an exception. Therefore, some explicit statements on this matter now appear in Section 4.3 and in Conclusion.
Reviewer 4 Report
In the revised version of the paper, the authors do not follow my reccommendations to shortening Introduction and Section 2.1 and exclude Section 4.2. In principle, although the paper is acceptable in the presented view but it spends unnecessarily much journal space. My advice is to abridge the text.
Author Response
I truly appreciate your concern regarding the volume of the paper. When reading, I personally prefer shorter and more concise text, too. However, I believe that this particular paper should address the background of very wide range of readers, starting from undergraduate engineers. Therefore, it should provide sufficient details and rationale to them.
In addition, other Reviewer's comments imply adding some more explanations to the text, which I am obliged to provide at this point.
"In the revised version of the paper, the authors do not follow my reccommendations to shortening Introduction and Section 2.1 and exclude Section 4.2"
This time, I have removed several sentences from Introduction and Section 2.1. In my opinion, more explanations are never worse. Some of them may really help to less experienced user, who may find our primary statements looking unsound.
In Section 2.1, three conclusions are being established to backup further derivations. For each of them, we have some textual explanation, some formulas, and graphical illustration. It is hard to substantially shrink the material without a risk of making our conclusions looking rather groundless for someone.
Removing Section 4.2 seem to contradict with the opinion of other Reviewer. They claim that method should be made simpler, without, however, providing any clue on in which part and how. Section 4.2 shows the drawbacks on one of plausible simplification by taking an ordinary linear model for thermal dynamics.